# Adjacency Search Embeddings

**Meher Chaitanya**                                                      *mpindiprolu@ethz.ch*
*ETH Zürich*

**Kshitijaa Jaglan**                                                      *kjaglan@ifi.uzh.ch*
*University of Zürich*

**Ulrik Brandes**                                                      *ubrandes@ethz.ch*
*ETH Zürich*

**Reviewed on OpenReview:** *https: // openreview. net/ forum? id= GDN5cFTNaL*

## Abstract

In this study, we propose two novel Adjacency Search Embeddings that are inspired by the theory of identifying *s-t* minimum cuts: Maximum Adjacency Search (MAS) and Threshold-based Adjacency Search (TAS), which leverage both the node and a subset of its neighborhood to discern a set of nodes well-integrated into higher-order network structures. This serves as context for generating higher-order representations. Our approaches, when used in conjunction with the skip-gram model, exhibit superior effectiveness in comparison to other shallow embedding techniques in tasks such as link prediction and node classification. By incorporating our mechanisms as a preprocessing technique, we show substantial improvements in node classification performance across GNNs like GCN, GraphSage, and Gatv2 on both attributed and non-attributed networks. Furthermore, we substantiate the applicability of our approaches, shedding light on their aptness for specific graph scenarios.

## 1 Introduction

Node embeddings are concise numerical representations of individual nodes in a network, capturing their structural and relational significance. These embeddings condense complex structural information into compact vectors for facilitating efficient downstream tasks such as node classification, link prediction, and other machine-learning applications. These embeddings place nodes in a continuous vector space, allowing us to extract meaningful features for tasks like node classification and link prediction in machine learning. Inspired by Natural Language Processing (NLP) techniques, node embedding methods like SkipGram-Word2Vec Mikolov et al. (2013b;a) use algorithms to capture intricate patterns and relationships within graph structures.

These NLP algorithms excel in generating embeddings that capture semantic relationships and contextual nuances within textual data. The SkipGram model focuses on predicting context words - those expected to appear in close proximity given a specified target word. Shallow embedding techniques for graphs generate sequences of nodes using methodologies like random walks Perozzi et al. (2014); Grover & Leskovec (2016) or auto-encoding adjacency vectors Wang et al. (2016) and employ the SkipGram model to generate vector representations. The resulting embeddings aim to encode each node in a graph into a low-dimensional vector, where the geometric proximity of vectors reflects the similarity or relatedness of the corresponding nodes in the network. By capturing both local and global graph structures, node embeddings prove instrumental in enhancing the efficiency and effectiveness of downstream machine learning tasks Xu (2021). These shallow embedding techniques, though computationally efficient, suffer from the drawback of lacking parameter sharing during training. Each node has a unique embedding vector that is learned independently. This lack of parameter sharing can lead to overfitting, especially when dealing with sparse graphs or limited training data Hamilton et al. (2017b). Recently, Graph Neural Networks (GNNs) have shown great promise in learning representations by sharing parameters across node neighborhoods for graph-structured data for

better generalization. However, training GNNs on large graphs can be computationally challenging due to the exponential growth of node neighborhoods with increasing network depth Hamilton et al. (2017a).

Neighbor sampling approaches have emerged as a key technique to address this issue by approximating the node neighborhoods during training. They enable scalable GNN training on large-scale graphs that would otherwise be intractable due to memory constraints. Early works like GraphSAGE Hamilton et al. (2017a) introduced neighborhood sampling as a way to make GNN training feasible on large graphs. Subsequent research has explored various sampling strategies, including layer-wise sampling, importance sampling, and variance reduction techniques Chen & Zhao (2018); Zou et al. (2019). Recent advancements like GraphSAINT Zeng et al. (2019) and ClusterGCN Chiang et al. (2019) have further demonstrated the effectiveness of neighbor sampling in achieving state-of-the-art performance on large-scale graph benchmarks. The continued development and adoption of neighbor sampling methods are crucial for unlocking the full potential of GNNs in real-world applications with massive graphs.

**Present work in a nutshell:** In this research, we introduce two methods, Maximum Adjacency Search (MAS) and Threshold-based Adjacency Search (TAS), aimed at capturing sequences of *well-connected* nodes from a given set of nodes. The TAS approach employs an extended version of the well-known Breadth First Search (BFS) Algorithm, called Delayed-BFS. This algorithm takes a set of vertices, $S$, and a threshold as input to generate a sequence of nodes that are structurally well-connected, potentially within a higher-order network topology, to $S$. These sequences can be fed to the SkipGram-Word2Vec model or can be used as a preprocessing or sampling mechanism in existing GNNs to identify structurally influential local neighborhoods for feature aggregation around a node to generate node embeddings. The MAS approach is a *dynamic* variant of TAS. Our goal in this paper is not to explicitly identify specific structures like motifs, graphlets, or simplicial complexes Vishwanathan et al. (2010); Rossi et al. (2018); Piaggesi et al. (2022) but rather generate sequences of tightly connected nodes in cohesive sub-structures of the network.

**Main Contributions:**

- **Novel sampling Approaches:** We propose two variants of Adjacency search-based methods, Maximum Adjacency Search(MAS) and Threshold-based Adjacency search (TAS) embeddings, which are efficient, effective, and highly scalable algorithms for feature learning on graphs. These approaches involve constructing sequences as neighborhood contexts, capturing nodes within the cohesive sub-structures in the network topology.

- **Non-random walk techniques:** Our approach involves executing Breadth-First Search (BFS) not from a single node but rather from a subset of nodes simultaneously based on certain threshold criteria to determine when a node can be marked as visited. This unique methodology enables the generation of node sequences that inherently capture complex topological patterns such as triads, quads, simplicial complexes, and motifs. Our path-based approaches differ from the random walk approaches and can be considered as specific instances of $k$-th order random walks for any $k \in \mathbb{N}$ [1].

- **Empirical Performance:** Our proposed algorithms significantly enhance link prediction accuracy when integrated with the SkipGram model, outperforming the best results of other shallow embedding techniques by an average of *6.97%*. Furthermore, our approach serves as an effective local neighborhood sampling technique for message aggregation in GNNs, leading to improved performance across various architectures, including GCN, GraphSage, and GATv2, for both attributed and non-attributed graphs. Notably, in non-attributed networks, our sampling mechanism demonstrates up to *12.1%* efficiency improvement compared to trainable node embeddings [2] in GNNs, highlighting its effectiveness in identifying structurally influential neighborhoods for aggregation.

---

[1] A higher-order random walk typically retains some memory of previous steps, as opposed to a first-order Markov process. While our approaches are not directly a memory-based walk in the strictest sense, its choice of nodes (based on maximum adjacency or threshold) implicitly reflects past choices. This makes them behave similarly to a higher-order random walk, where past decisions influence the current selection

[2] trainable node embeddings refer to the node embeddings which can be initialized with any randomly-initialization methods (https://pytorch.org/docs/stable/generated/torch.nn.Embedding.html)

- **Applicability:** We discuss the applicability of our approach and identify scenarios where it may not be well-suited for downstream tasks in Appendix D.

**Preliminaries**  Here, we present a brief overview of Maximum Adjacency Search (MAS), an effective method for determining *s-t* minimum cuts. MAS Stoer & Wagner (1997) is a widely used algorithm in graph theory, particularly for identifying minimum cuts and analyzing flow networks. In the context of an *s-t* minimum cut, the goal is to partition the graph into two disjoint sets: one containing the source vertex *s*, and the other containing the target vertex *t*. The objective is to minimize the total weight of the edges that connect these two sets. MAS works by selecting vertices based on their adjacency to already-selected vertices. Specifically, in each iteration, a vertex *v* with the highest adjacency among the unselected vertices is added to the current set. This iterative process continues greedily, including the vertex with the most adjacent neighbors already in the set and ensuring that critical vertices and edges are efficiently identified for the partition. This heuristic not only facilitates efficient identification of the minimum cut but also tends to form tightly connected communities around the seed set *S*. The likelihood of selecting a cut-edge is minimized at each iteration by choosing a highly connected vertex, which helps avoid cutting through critical edges prematurely and promotes the formation of cohesive vertex groupings.

**Motivation**  Our approaches are motivated by Maximum Adjacency Search, which is a well-known *s-t* minimum cut approach. They are built on the assumption that the probability of future connections between two different nodes is higher in well-connected structures. Thus our methods are based on the assumption that the likelihood of future connections between two distinct nodes is higher within *well-connected* substructures. Unlike random walks, which sample sequences across the entire graph, our approaches leverage MAS and its generalized variants to generate targeted sequences that capture *well-connected* local neighborhoods around each node. This approach reduces training time, as fewer sequences are generated compared to random walk-based methods. Furthermore, in contrast to existing higher-order techniques, our methods efficiently identify these well-connected regions without explicitly searching for combinatorial structures, such as cliques.

## 2   Related Work

Over the years, various algorithms and techniques have been proposed for generating node embeddings, each with its strengths and applications. These can be broadly classified into three major categories: Matrix Factorization Approaches, Random Walk Approaches, and Graph Neural Networks (GNNs). This section briefly overviews various embedding approaches specifically tailored for non-attributed graphs.

**Matrix Factorization Approaches:**  Early techniques for learning representations through matrix factorizations draw significant inspiration from dimensionality reduction methods Belkin & Niyogi (2001); Kruskal (1964). Methods falling within this category construct a high-order proximity matrix using transition probabilities and subsequently engage in factorization processes to derive node embeddings. Notable approaches, such as Laplacian Eigen Maps Belkin & Niyogi (2001) and Locally Linear Embeddings (LLE) Rowes (2000), operate based on this foundational principle. HOPE (High-Order Proximity preserved Embedding) Ou et al. (2016) strives to maintain higher-order proximity information within the learned embeddings. It achieves this objective by performing a matrix factorization strategy to capture higher-order proximities in the form of a similarity matrix. By decomposing this matrix, HOPE generates embeddings that preserve first and higher-order relationships within the graph. The HOPE algorithm supports general similarity measures, distinguishing it from other approaches. For instance, the Graph Factorization algorithm Ahmed et al. (2013) considers the first-order neighborhood as a similarity measure, while GraRep Cao et al. (2015) operates with higher-order similarity, such as $k$-th length shortest paths between two nodes ($A^k$). It's important to note that these approaches may not be suitable for large graphs because their loss functions rely on the entire adjacency matrix Xu (2021).

**Random Walk Approaches:**  A significant number of recent successful methodologies utilize random walk statistics to learn node embeddings. The underlying concept is to ensure similar embeddings for nodes that co-occur during random walks Goyal & Ferrara (2018). The DeepWalk algorithm Perozzi et al. (2014), for instance, utilizes random walks to explore the graph and subsequently applies skip-gram or continuous

bag-of-words models for generating embeddings. Node2Vec Grover & Leskovec (2016) extends DeepWalk by incorporating second-order random walks. Notably, Node2Vec introduces flexible parameters, the "return parameter" and the "in-out parameter," allowing for a trade-off between exploration and exploitation during the generation of sentences of nodes through random walks. Fine-tuning these parameters enables Node2Vec to produce embeddings that emphasize either local neighborhood structures (homogeneous neighborhoods) or more extensive structural information (heterogeneous neighborhoods). Various extensions to these shallow embedding methods have been proposed in the literature Chen et al. (2018a); Perozzi et al. (2017); Chamberlain et al. (2017). LINE Tang et al. (2015) is another approach often compared with the performance of DeepWalk and Node2Vec. LINE aims to discover a $d$-dimensional representation for each node by considering first and second-order graph proximities. It is similar to DeepWalk when the size of the vertices context is set to one Qiu et al. (2018). Recently, Pimentel et al. (2019) proposed neighborhood-based node embeddings (NBNE) that used ego-centric representations of nodes instead of random walks for learning node representations. Their approach attains superior performance compared to Node2Vec and SDNE Wang et al. (2016) on prediction and classification tasks. The approaches Node2Vec, DeepWalk, and NBNE can be considered neighborhood search-based embedding approaches for a node.

Further related work pertinent to GNNs tailored for non-attributed networks is provided in Section 5.1.

## 3    Adjacency Search Based Embeddings

Within graphs, approximating a node's context often involves constructing or sampling sequences. For example, Node2Vec employs second-order biased random walks Grover & Leskovec (2016), while DeepWalk utilizes uniform random walks Perozzi et al. (2014) for generating these sequences. In this section, we introduce two novel mechanisms for generating sequences to serve as contexts for nodes. Before delving into our approaches, we present the relevant notation used in this paper. We represent the graph by $G(V, E)$ where $V$ and $E$ are the sets of vertices and edges, respectively. For ease of notation, let $n$ and $m$ represent the cardinality of the vertex and edge sets, respectively. The open and closed neighborhoods of a vertex $v$ are denoted by $N(v)$ and $N[v]$. We denote the set of positive integers by $\mathbb{Z}^+$. We use node and vertex interchangeably in this paper.

### 3.1    Maximum Adjacency Search (MAS)-Embeddings

This approach draws inspiration from the Maximum Adjacency/Cardinality search technique, which is employed for identifying an $s$-$t$ minimum cut in a graph Stoer & Wagner (1997); Cai & Matula (1993). It is a recursive algorithm where, at each iteration, the objective is to identify a tightly connected vertex to the set of well-connected vertices and perform a merge operation to identify the $s$-$t$ minimum cut. Incorporating the vertex with the highest degree of connectivity into the existing set of vertices reduces the likelihood of encountering an edge belonging to the minimum cut.

Our MAS-Embeddings approach provided in Algorithm 1 works similarly by constructing sequences of *well-connected* and *cohesive* nodes for every vertex $v$, together with a subset of its neighborhood, $S \in N[v]$. This is attained by iteratively adding a vertex $u \in V$ that is tightly connected (maximally connected) to the sequence set $S$ (line 11 in Algorithm 1). This process is repeated until the desired Walk Length is attained. The MAS-Embeddings approach relies on the following parameters (all the parameters are in $\mathbb{Z}^+$):

- *Number of Neighbors* ($k$): $k$ denotes the cardinality of the selected neighborhood of $v \in V$.
- *Number of Permutations* ($p$): This parameter assists in capturing diversified immediate neighborhoods of a vertex $v$, which in turn facilitates the identification of a cohesive subset of nodes.
- *Walk Length* ($l$): This parameter refers to the sequence length that is generated for every vertex in the graph.
- *Window Size* ($w$): This refers to the context window size used by the SkipGram model.
- *Dimensions* ($d$): This parameter represents the dimension of feature representations.

---

**Algorithm 1** MAS-Embeddings

---

1: **Input:** Graph $G$, Walk Length $l$, Window size $w$, Dimension $d$, #Neighbors $k$, #Permutations $p$
2: **for all** $v$ in graph.$nodes()$ **do**
3:      **repeat** $p$ **times**
4:          $neighbors \leftarrow$ permute($v$.neighbors())          ▷ random permutation of the neighbors of a vertex
5:          $num\_sequences \leftarrow$ len($neighbors$)/$k$
6:          **for** $i \leftarrow 1....num\_sequences$ **do**
7:              $S \leftarrow \varnothing$
8:              $S$.insert($v$)
9:              $S$.insert($neighbors[l \cdot k...l \cdot (k+1)]$)          ▷ insert $k$ neighbors of node $v$ into the set $S$
10:              **repeat**
11:                  find the maximally adjacent $u$ to $S$
12:                  $S$.insert($u$)
13:              **until** (len($S$) = $l$)
14:              $sequences$.insert($S$)
15:      **end**
16: $f =$ train($w, d, sequences$)          ▷ train the sequences using skip-gram model
17: **return** $f$

---

## 3.2 Threshold-based Adjacency Search (TAS)-Embeddings

Here, we present the TAS embeddings, a generalized variant of the MAS embeddings approach. Let $\theta \in \mathbb{R}^n$ be a vector of thresholds, where each element $\theta_v$ denotes the threshold for vertex $v \in V$, and $\theta_v \in \mathbb{Z}+$. The transition state of a vertex $v$ from non-visited to visited occurs when at least $\theta_v$ of its neighbors are already visited. A sequence for node $v$ is the set of vertices visited by $v$ together with a subset of its neighbors $S \subseteq N[v]$. For the rest of the paper, we consider $\theta$ to be a constant, i.e., all vertices have uniform thresholds.

### 3.2.1 Generation of Sequences

The Delayed-BFS approach, outlined in Algorithm 3, takes a vertex $v$, a subset of its neighborhood $S \in N[v]$, and threshold $\theta$ as the input to produce a sequence of nodes that are cohesive and well-connected to $S$. The thresholds of all $u \in V \setminus S$ are set to the *minimum* of their degree and $\theta$. This is because the maximum threshold at which $u$ can be visited is $\theta_u = $ degree($u$). The threshold parameter ensures that the vertex visit is delayed till it acquires sufficient reinforcement from its neighbors. When $\theta = 1$, Delayed-BFS($S$) behaves similarly to traditional BFS($S$), encompassing immediate neighbors and their subsequent neighbors in the sequence. The distinction from traditional BFS becomes apparent when $\theta \geq 2$. For $\theta = 2$, every vertex needs reinforcement from at least 2 visited neighbors for it to be visited. To illustrate the operation of Algorithm 3, consider Figure 1, where for $\theta = 3$, Delayed-BFS($1, 2, 3$) does not visit any nodes. In contrast, Delayed-BFS($2, 4, 6$) will visit $\{5, 3, 7\}$. Similarly, Delayed-BFS($1, 3$) with $\theta = 2$ will visit the set of vertices $\{2, 4, 5, 6, 7, 8, 9\}$. The set of sequences of nodes generated for different thresholds adheres to the following hierarchy:

$$\text{sequence}_{\theta=1} \supseteq \text{sequence}_{\theta=2} \supseteq \text{sequence}_{\theta=3} \cdots$$

Thus, the generated sequences are constrained to visit specific cohesive structures (including higher-order structures such as motifs, etc) with an increase in thresholds $\theta$. Notice that the MAS Embeddings approach (Algorithm 1) can be interpreted as a *dynamic* threshold-based adjacency search embedding where the threshold of the node $u$ that is being added to the sequence $S$ is equal to the cardinality of the set of edges between $u$ and $S$, i.e., $|E(u, S)|$. As illustrated, TAS-Embeddings (Algorithm 2) produces distinct sequences by employing the Delayed-BFS approach for various subsets of a node and its neighborhood. Apart from the set of parameters mentioned in Section 3.1, the Algorithm 2 relies on the threshold parameter, which is a constant in our experiments.

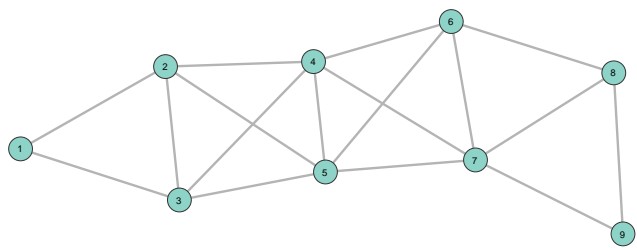

Figure 1: An example network to demonstrate the functioning of Delayed-BFS approach (3)

---

**Algorithm 2** TAS-Embeddings
---
1: **Input:** Graph $G$, Threshold $\theta$, Walk Length $l$, Window size $w$, Dimension $d$, #Neighbors $k$, #Permutations $p$
2: **for all** $v$ in graph.$nodes()$ **do**
3:      **for** threshold $\leftarrow 1...\theta$ **do**
4:          **repeat** $p$ **times**
5:             $neighbors \leftarrow \text{permute}(v.\text{neighbors}())$        ▷ random permutation of the neighbors of a vertex
6:             $num\_sequences \leftarrow \text{len}(neighbors)/k$
7:             **for** $i \leftarrow 1...num\_sequences$ **do**
8:                 $S \leftarrow \varnothing$
9:                 $S.\text{insert}(v)$
10:                $S.\text{insert}(neighbors[l \cdot k...l \cdot (k+1)])$       ▷ insert $k$ neighbors of node $v$ into the set $S$
11:                $sequence \leftarrow \text{Delayed-BFS}(G, S, \theta, l)$        ▷ generate the threshold based sequences
12:                $sequences.\text{insert}(sequence)$
13:          **end**
14: $f = \text{train}(w, d, sequences)$        ▷ train the sequences using skip-gram model
15: **return** $f$

---

**Algorithm 3** Delayed-BFS
---
1: **Input:** Graph $G$, sequence $S$, threshold $\theta$, Walk Length $l$
2: **Initialization:** Queue $I$; $\forall u \notin S : T[u] = \min(degree(u), \theta)$; $T[v \in S] = 0$
3: $\text{enqueue}(I, S)$
4: **repeat**
5:      vertex $i \leftarrow \text{dequeue}(I)$
6:      $sequence.\text{insert}(i)$
7:      **for all** $u \in N[i]$ **do**
8:          **if** $u \notin I$ **then**
9:             decrement $T[u]$
10:             **if** $T[u] == 0$ **then**
11:                $\text{enqueue}(I, u)$
12: **until** ($I$ is empty) or ($\text{len}(sequence) = l$)
13: **return** $sequence$

---

### 3.3 Shallow Embedding Learning Framework

Let $f : V \to \mathbb{R}^d$ be the function that maps nodes to d-dimensional feature representations that we intend to learn for downstream tasks. Let $\mathcal{S}$ denote the set of sequences generated by Algorithm 2. We adopt the standard feature learning framework where the representations are trained by maximizing the log probability of predicting a node occurrence in a sequence $s \in \mathcal{S}$ conditional on another node and its representation

Hamilton et al. (2017b). This is expressed as:

$$\max_f \frac{1}{|\mathcal{S}|} \sum_{v \in V} \log \Pr(s|f(v)) \tag{1}$$

where Pr is the probability measure. under the standard assumptions of conditional independence to ensure tractability, we get

$$\Pr(s|f(v)) = \sum_{u \in s} \Pr(u|f(v))$$

Modeling the conditional likelihood as softmax parametrized by feature vectors:

$$Pr(i \in s|f(v)) = \frac{e^{\langle f(i), f(v) \rangle}}{\sum_{u \in V \setminus v} e^{\langle f(u), f(v) \rangle}}$$

where $\langle f(i), f(v) \rangle$ represents the dot product between the feature representation of nodes $i$ and $v$. By optimizing the log probability equation 1, the algorithm maximizes the predictability of a node $i$ in the generated sequence given another node $j$, thereby creating node embeddings in which nodes co-occurring in sequences have similar representations. This optimization is accomplished using stochastic gradient descent with negative sampling Mikolov et al. (2013b).

### 3.4   Time Complexity

In this section, we assess the time complexity of our proposed embedding approaches. Consider a graph $G$ with $n$ and $m$ representing the cardinalities of the vertex and edge sets. Let $E(v)$ denote the cardinality of the set of edges associated with the vertex $v$. The maximum degree in the graph is denoted by $\Delta$. Let $S$ be the set comprising of the node and a subset of its neighborhood for which a sequence is generated (Algorithm 1). We create $\Delta$ buckets where vertices are distributed in these buckets according to the count of their adjacencies with $N[S]$. Notice that, under this construction, adding a vertex $u$ to set $S$ necessitates at most $E(u)$ updates (moving a vertex from one bucket to another) and identifying the next maximum adjacent vertex to $N[S]$ involves a constant look-up (from the highest bucket). The time complexity for generating $\mathcal{O}(n)$ sequences using Algorithm 1 is given by:

$$\mathcal{O}\Big(\sum_{i=1}^{n}\Big(\sum_{x=1}^{p}\Big(\sum_{m=1}^{\frac{deg(v)}{k}}\Big[E(S) + \sum_{j=1}^{l-|S|} E(u_j)\Big]\Big)\Big)\Big) \tag{2}$$

where $u_j$ is the $j^{th}$ vertex added to the set $S$ (sequence). The equation 2 evaluates to $\mathcal{O}(n + m)$ (since $\{l, p, k\} \in \mathcal{O}(1)$). The time complexity of SkipGram-Word2Vec is linear on the number of sentences and logarithmic on the size of vocabulary (i.e., on the vertex set $|V| = n$) Mikolov et al. (2013a); Pimentel et al. (2018). Thus, the overall time complexity of the Maximum Adjacency Search embeddings (Algorithm 1) is bounded by $\mathcal{O}((n + m) \log n)$. A similar analysis applies to Threshold-based Adjacency Search Embeddings (see Algorithm 2). In practice, MAS outperforms TAS in computational efficiency due to its significantly smaller set of hyperparameters, which results in a reduced number of generated sequences. Since walk lengths are bounded by a constant, the space complexity is linear in the number of vertices. Further details on the scalability of MAS and TAS approaches are provided in the Appendix A.

## 4   Empirical Evaluation with other Shallow Embedding Techniques

We assess the feature representations derived from MAS and TAS embeddings in the context of standard supervised learning tasks: link prediction and node classification. We utilize six datasets to evaluate the performance of our approaches against other shallow embedding methods. The statistics of these six graph datasets employed for evaluation are outlined in the Appendix B, Table 8.

**Baselines:** We assess the efficacy of our proposed approaches in comparison to shallow embedding approaches like Node2Vec, DeepWalk, and NBNE (Results regarding the comparison with GNNs are detailed in Section 6). As outlined below, these methods are executed with a varied range of hyperparameters (with Dimensions, $d$, uniformly set to 128 for all approaches). The best-performing baseline is used for comparison.

- Node2Vec Grover & Leskovec (2016): Hyperparameter tuning systematically explores parameter values within a designated grid. Specifically, the parameters $p$ and $q$ undergo optimization on the validation set through an exhaustive grid search, considering values from the set $[0.1, 0.2, \ldots, 1, 2, 4]$. The window size, denoted as $l$, is selected from the set of values $\{5, 10, 15, 20\}$. The walk lengths are chosen from the range $[10, 80]$ in increments of 10, and the number of random walks per node is set to 10.

- DeepWalk Perozzi et al. (2014): The sampling strategy can be seen as a special case of node2vec with $p = q = 1$. The rest of the parameter configurations are similar to Node2Vec.

- NBNE Pimentel et al. (2019): NBNE uses a permutation parameter ($\phi$) to capture diversified immediate neighborhoods. we vary $\phi$ from 1 to 20, and the window size is set to the subset of the neighborhood (varied from 3 to 10) that is chosen for representation (as suggested in the paper).

**Parameters for MAS and TAS:** The parameters for MAS and TAS described in the Algorithms 1 and 2 are initialized to the following values.

- *Number of Neighbors* ($k$): In our experiments, we vary $k$ from 1 to 10. For graphs having higher average degrees (such as Facebook and email), we vary the range of *number of neighbors* ($k$) parameter from 1 to 20.
- *Number of Permutations* ($p$): In our experiments, $p$ is a small constant, and we vary it from 1 to 8.
- *Walk Length* ($l$): In our experiments, we vary this parameter from 10 to 40 in steps of 5.
- *Window Size* ($w$): This is chosen from the set of values $\{5, 10, 15, 20\}$.
- *Dimensions* ($d$): For experimental purposes, we set $d = 128$.
- Threshold ($\theta$): For our empirical analysis, the threshold parameter, $\theta \in \mathbb{Z}^+$, is varied in the range of $\theta \in [1, 8]$.

**Link Prediction Task:** In the context of a network, link prediction tasks involve predicting the likelihood of the existence of a link (edges) between pairs of nodes based on the observed network topology and associated features Lü & Zhou (2011). Node embeddings are extended to edge embeddings using the operators Hadamard ($\odot$), $l2$, $l1$, and Average. Further description of these operators is provided in the Appendix, Table 9. Subsequently, a logistic classifier is trained on these edge embeddings to identify the potential presence of an edge in the network. For training both Maximum Adjacency Search (MAS) and Threshold-based Adjacency Search (TAS) embeddings, we initially obtain a sub-graph with 80% randomly selected edges from each dataset and generate node embeddings by training MAS and TAS on these sub-graphs. A subset of the training edges (10%) are used for validation to select the best operator. The best logistic regression classifier obtained in this process is further used for testing on the remaining 20% data. We use an equal-sized sample of negative edges as positive edges for training. The accuracy for link prediction tasks is evaluated using the AUC (area under the ROC curve) score with a 5-fold cross-validation Baeza-Yates (1999).

Table 1 delineates the performance of Maximum Adjacency Search (MAS) embeddings and Threshold-based Adjacency Search (TAS) embeddings in link prediction tasks. On average, MAS and TAS approaches demonstrate an improvement in accuracy of 6.075% and 6.76%, respectively, surpassing the best accuracy achieved by other methods. Notably, the Hadamard operator (denoted by $\odot$ in Table 1), when used with MAS and TAS embedding approaches, exhibits good stability and consistently yields the best performance across all networks. We provide the percentage gain attained by MAS and TAS over other approaches in Table 1.

Table 1: AUC of Maximum Adjacency Search (MAS) Embeddings, Threshold-based Adjacency (TAS) embeddings, Node2vec, Deepwalk, and NBBE on Link Prediction task. On all the datasets, the $\odot$ operator achieved the best performance. The best overall performance is highlighted in bold, while the best-performing baseline is shaded in gray.

| Data set | MAS Embeddings | TAS Embeddings | Node2vec | DeepWalk | NBNE | Gain |
|---|---|---|---|---|---|---|
| FaceBook | 0.985 | **0.988** | 0.933 | 0.927 | 0.921 | 5.89% |
| arXiv | 0.983 | **0.985** | 0.923 | 0.911 | 0.927 | 6.25% |
| Email | **0.941** | 0.932 | 0.857 | 0.829 | 0.843 | 9.80% |
| Cora | 0.868 | **0.881** | 0.838 | 0.749 | 0.842 | 4.6% |
| Pubmed | 0.885 | **0.913** | 0.801 | 0.783 | 0.838 | 8.95% |
| Citeseer | **0.871** | 0.869 | 0.761 | 0.729 | 0.819 | 6.35% |

**Node Classification:** Node classification is the task of assigning predefined labels or categories to nodes within a network. Each node is associated with one or more classes, and machine learning models utilize node features, network topology, or a combination of both to predict these labels Goyal & Ferrara (2018). In our study, we utilize node embeddings generated through Maximum Adjacency Search (MAS) and Threshold-based Adjacency Search (TAS) approaches across the entire network. The process involves training logistic classifiers with 80% of the labeled nodes for training, including a subset for validation, while the remaining 20% serves as a test set. This procedure is iterated with 5 random seed initializations to ensure the robustness and reliability of the evaluation process. Table 2 shows the Micro-F1 scores for the node classification task on datasets Email, Cora, Pubmed, and Citeseer. As can be observed, both MAS and TAS embedding approaches considerably outperform the existing shallow embedding methodologies by an average improvement of 15.95% and 19.67%, respectively. All datasets are used for multi-class classification except for the Blog dataset, which uses multilabel classification.

Table 2: Micro-F1 score of MAS embeddings, TAS embeddings, Node2vec, Deepwalk and NBNE on node classification task. The best overall performance is highlighted in bold, while the best-performing baseline is shaded in gray.

| Data set | MAS Embeddings | TAS Embeddings | Node2vec | DeepWalk | NBNE | Gain |
|---|---|---|---|---|---|---|
| Email | 0.1162 | **0.1263** | 0.0912 | 0.0733 | 0.0872 | 38.4 % |
| Cora | **0.8271** | 0.8192 | 0.7530 | 0.7427 | 0.7701 | 7.4 % |
| Pubmed | 0.7931 | **0.8282** | 0.7181 | 0.7007 | 0.6914 | 15.33% |
| Citeseer | 0.6294 | **0.6296** | 0.5013 | 0.5071 | 0.5309 | 18.59% |
| Blog | **0.2103** | 0.2097 | 0.1731 | 0.1514 | 0.1901 | 10.62% |

## 5 Neighborhood Sampling in GNNs

Neighborhood sampling in Graph Neural Networks is a technique designed to manage computational complexity and memory usage by selectively sampling a subset of a node's neighbors during training. Traditional GNNs, like Graph Convolutional Networks (GCNs) Kipf & Welling (2016a), aggregate information from all neighboring nodes, which can be computationally prohibitive for large-scale graphs. To address this, methods such as GraphSAGE (Graph Sample and AggregatE) Hamilton et al. (2017a) and FastGCN (Fast Graph Convolutional Networks) Chen et al. (2018b) employ neighborhood sampling to limit the number of neighbors considered at each layer. GraphSAGE samples a fixed number of neighbors for each node to create a computationally feasible and efficient aggregation process, enabling the model to handle large graphs without significant loss in performance. FastGCN, on the other hand, samples neighbors in a way that approximates the full distribution of the graph, thereby maintaining the quality of the learned representations while reducing

the computational load. Other notable approaches include importance sampling, where neighbors are selected based on their relevance, adaptive sampling techniques that adjust the sampling strategy during training and training on smaller sub-graphs or clusters Chen & Zhao (2018); Zou et al. (2019); Huang et al. (2018); Chiang et al. (2019); Zeng et al. (2019). These sampling strategies are critical for scaling GNNs to industrial-sized graphs used in real-world applications such as social networks, recommender systems, and biological networks.

### 5.1 Literature Review

Graph Neural Networks (GNNs) employ edge-based aggregation of node features into low-dimensional representations, demonstrating effectiveness in various downstream machine learning tasks Izadi et al. (2020); Kan et al. (2021); Kipf & Welling (2016b); Zhang & Chen (2018); Zhang et al. (2018). Nevertheless, the application of many existing GNNs is limited to graphs with node features/attributes. A considerable number of real-world graphs do not include node attributes Chen et al. (2020); Duong et al. (2019). In such scenarios, the direct application of Graph Neural Networks (GNNs) becomes challenging due to the absence of node features Cai & Wang (2018); Errica et al. (2019). Various intuitive methods have been commonly employed to address this issue by selecting artificial node features for GNNs based on degree, random, positional, and distance-related characteristics Hamilton et al. (2017a); Sato et al. (2021); Chen et al. (2017); You et al. (2019); Li et al. (2020). Recently, Cui et al. (2022) provided a comprehensive understanding of artificial node features for positional node classification tasks. The authors categorize commonly used artificial node features into positional and structural node features based on the type of information they can assist GNNs in capturing. Empirical results indicate that positional node features are particularly effective for positional node classification, while structural node features prove more beneficial for tasks related to structural node classification and graph classification. The authors observe that using DeepWalk embeddings as positional node features for GNNs yields the highest accuracy in node classification tasks compared to other feature alternatives such as random initialization (trainable node embeddings), degree, PageRank, etc.

Many structural encoding techniques have been proposed to address the limitations of GNNs to capture the underlying structure. In Bouritsas et al. (2022), the authors introduce a novel topologically-aware message passing scheme that leverages substructure encoding. The central concept involves counting specific substructures, such as cycles, cliques, and triangles, and embedding this substructure count information into the message-passing process of the neural network. Experimental results demonstrate that this method outperforms existing approaches. However, a significant limitation of such structure-encoding methods is the high time complexity ($\mathcal{O}(n^k)$ for a size $k$ substructure) associated with counting substructures. In Yan et al. (2024), the authors propose a technique for precomputing structural embeddings that encode distance information within simple subgraphs or substructures surrounding each node or an edge. This method enhances standard Graph Neural Networks (GNNs) by integrating precomputed topological information, which eliminates the need for the model to learn representations over all possible subgraphs dynamically during training. While the primary goal of these frameworks is to enhance the expressiveness of graph neural networks by explicitly counting these substructures, the Maximum Adjacency Search (MAS) and Topological Adjacency Search (TAS) frameworks proposed in Algorithms 1, 2 offer an efficient algorithmic procedure to implicitly capture well-connected substructures without the need to explicitly identify or count them, thus avoiding the computational bottleneck typically seen in substructure counting. This implicit approach enables faster and more scalable graph learning, while still improving the model's expressivity.

Path-based Graph Neural Networks inherently aims to capture the substructures by sampling random or shortest paths that extends beyond the simple neighborhood of the node. In Abboud et al. (2022), the authors introduces a novel message-passing mechanism for Graph Neural Networks (GNNs) that incorporates the shortest-path distance between nodes to address the issue of over-squashing. Traditional GNNs only allow nodes to communicate with their direct neighbors, which can limit the flow of information across the graph. To mitigate this, the proposed approach extends message passing beyond immediate neighbors by also considering $k$-hop nodes (nodes that are $k$ steps away) and their respective distances. This allows nodes to directly exchange information even if they are not adjacent, thereby breaking the information bottleneck. PathGCN Eliasof et al. (2022) employs simple random walks as paths over which the weights for the spatial operator are learned. These random walks define the paths across which information is propagated. After learning the weights associated with these paths, a convolution operation is performed using a linear operator.

Path Neural Networks (PathNNs) Michel et al. (2023) are a novel graph neural network architecture that enhances the expressive power of GNNs by aggregating information along paths emanating from each node. PathNNs aggregate paths of varying lengths, such as shortest paths or simple paths up to a certain length $K$, to update node representations. Unlike random walks Eliasof et al. (2022) or shortest paths Abboud et al. (2022); Michel et al. (2023), which sample connected sequences from each node, our MAS and TAS approaches generate targeted sequences that capture well-connected local paths around each node. These paths can be seen as instances of higher-order random walks and represent a distinct form of complex shortest paths, facilitated by the Delayed-BFS algorithm outlined in Algorithm 3. This approach not only reduces training time by generating fewer sequences compared to random walks but also enhances the model's expressivity by capturing more intricate structures within the graph.

## 5.2 Threshold Adjacency Search (TAS) neighborhood Sampler

In this section, we introduce *TAS-sampler*, a novel sampling technique designed to identify structurally significant local neighborhoods surrounding nodes within Graph Neural Networks for aggregation and message passing. This technique can be seamlessly integrated with any existing GNN architecture. To showcase the effectiveness of *TAS-sampler*, we incorporate it into three prominent GNN architectures for non-attributed graphs: GCN Kipf & Welling (2016a), GraphSage-Pool Hamilton et al. (2017a), and GATv2 Brody et al. (2021).

Algorithm 4 outlines the *TAS-Sampler* procedure. First, the optimal parameters for the threshold, the number of neighbors, and the number of permutations are determined using Algorithm 2. Then, sequences representing the node's neighborhood context are generated, and an auxiliary graph, $G'$, is constructed using this new context. While MAS could be used for sequence sampling, we opted for TAS sequences due to their strong performance in node classification tasks (Table 2).

---

**Algorithm 4** TAS-Sampling

1: **Input:** Graph $G$
2: $\{\theta, k, p\}$ = TAS-Embeddings()                           $\triangleright$ get the optimal parameters for TAS
3: **for all** $v$ in graph.*nodes*() **do**
4:     **for** threshold $\leftarrow 1...\theta$ **do**
5:         **repeat** $p$ **times**
6:             $neighbors \leftarrow$ permute($v$.neighbors())     $\triangleright$ random permutation of the neighbors of a vertex $v$
7:             $num\_sequences \leftarrow$ len($neighbors$)$/k$
8:             **for** $i \leftarrow 1...num\_sequences$ **do**
9:                 $S \leftarrow \varnothing$
10:                $S$.insert($v$)
11:                $S$.insert($neighbors[l \cdot k...l \cdot (k+1)]$)         $\triangleright$ insert $k$ neighbors of node $v$ into the set $S$
12:                $sequence \leftarrow$ Delayed-BFS($G, S, \theta, l$)
13:                $sequences$.insert($sequence$)
14:         **end**
15:     TAS_Adjacency_List($v$) = $H$ most frequent vertices from $sequences(v)$.
16: Construct an auxiliary graph $G'$ with TAS_Adjacency_List
17: **return** $G'$

---

During each message passing iteration, the hidden embedding $h_u^{(k)}$ corresponding to node $u \in V(G')$ is updated according to the information aggregated from $u's$ neighborhood $N(u) \in G'$. This is expressed as follows (For consistency, we use the similar notation from Hamilton (2020)):

$$\mathbf{h}_u^{(k+1)} = UPDATE^{(k)}\Big(\mathbf{h}_u^{(k)}, AGGREGATE(\{\mathbf{h}_v^{(k)}, \forall v \in N(G'(u))\})\Big) = UPDATE^{(k)}\Big(\mathbf{h}_v^k, \mathbf{m}_{N(G'(u))}^{(k)}\Big)$$

where $UPDATE$ and $AGGREGATE$ are arbitary differentiable functions (neural networks), $V(G')$ are the set of vertices in the auxiliary graph $G'$ (step 17 of Algorithm 4), $N(G'(u))$ is the neighborhood of $u$ in $G'$ and

$\mathbf{m}_{N(G'(u))}$ is the aggregated message from the neighborhood of $u$ in $G'$. Observe that not all neighbors of $u$ in the original graph $G$ are retained as neighbors in the auxiliary graph $G'$, effectively creating skip-connections. Moreover, nodes well-connected to u but at a multi-hop distance in $G$ can become direct neighbors in $G'$.

## 6 Empirical Evaluation on GNNs with Artificial Node Features

Here we compare the performance of *TAS-sampler* incorporated in GNNs against *complete neighborhood*, *random sampling*, and *PageRank* sampling techniques Hamilton et al. (2017a) with artificial node features for node classification tasks. We employ a similar setup of experiments for node classification tasks as done in the paper Cui et al. (2022). We used the GraphSAGE, GCN, and GATv2 models, trained and tested them using the same data splits as in previous studies Kipf & Welling (2016a); Brody et al. (2021); Cui et al. (2022), namely 20 randomly selected samples for each class during training with a validation set of 500 samples. Trainable node embeddings [3] and the embeddings produced from DeepWalk are provided as artificial node features for GNN. In the case of trainable node embeddings, The performance is measured for mean, sum, and max aggregation in each GNN layer, and the best classification accuracy is reported in Table 3. For GCN and GATv2, we consider the complete neighborhoods of the original graph $G$ and TAS-sampled auxiliary graph $G'$ (step 16 of algorithm 4) for aggregation, and for GraphSage, we show the best performance attained by using random sampling and sampling using random walks with restarts (*PageRank*). For our experimental purposes, we initialized $H = 10$ (step 16 of Algorithm 4). The experiment is repeated for 5 random seed initializations for reliability. We also provide the accuracies obtained on Cora, Pubmed, and Citeseer datasets using GNNs with real node features. As can be observed from Tables 3, 4, 5, *TAS-Sampler* when incorporated in existing architectures, achieve an improvement up to 12.1% over other GNN architectures on these benchmark datasets showcasing its capability to identify influential neighbors for aggregations compared to existing mechanisms.

Table 3: F1-micro Accuracies on Cora Dataset for different initializations of features vectors for the node classification task. The highest overall performance is highlighted in bold, while the best-performing baseline is shaded in gray.

| GNN-Architecture | Trainable Node Embeddings | Deep walk Embeddings | real features |
|---|---|---|---|
| GCN | $59.7 \pm 1.1$ | $68.3 \pm 1.2$ | $76.4 \pm 1.3$ |
| GraphSage-pool | $63.8 \pm 1.7$ | $72.6 \pm 1.5$ | $77.7 \pm 1.1$ |
| GATv2 | $61.7 \pm 3.2$ | $71.7 \pm 1.3$ | $78.3 \pm 2.1$ |
| GCN-*TAS* | $\mathbf{71.8 \pm 2.1}$ | $73.7 \pm 1.5$ | $77.3 \pm 1.7$ |
| GraphSage-pool-*TAS* | $70.2 \pm 2.1$ | $\mathbf{74.2 \pm 1.9}$ | $78.2 \pm 0.9$ |
| GATv2-*TAS* | $68.1 \pm 1.4$ | $73.1 \pm 0.6$ | $\mathbf{79.4 \pm 1.6}$ |

Integrating the *TAS-Sampler* into existing GNN architectures yielded a substantial performance boost in non-attributed networks, both for trainable node embeddings and deep walk features. A reasonable performance increase was also observed for real node features, except for the Citeseer dataset, which has an average degree of 1.42 and an average clustering coefficient of 0.14. Due to the scarcity of higher-order structures in Citeseer, the *TAS-Sampler*, which samples nodes from cohesive regions, provides only marginal improvement on this dataset for real-world features. This behavior aligns with our discussion in Section D, as many nodes in extremely sparse graphs tend to have high betweenness centralities due to the lack of cohesiveness.

**Subgraph Sampling to find the optimal TAS parameter set (step 2 of Algorithm 4):** For all the experiments listed above, the optimal parameter set, $\{\theta, k, p\}$ is found by running *TAS* on the entire network $G$. Here, we study the perturbation of the node classification accuracies using GCN Kipf & Welling

---

[3]trainable node embeddings refer to the node embeddings which can be initialized with any randomly-initialization methods (https://pytorch.org/docs/stable/generated/torch.nn.Embedding.html)

Table 4: F1-micro Accuracies on CiteSeer Dataset for different initializations of features vectors for the node classification task. The best overall performance is highlighted in bold, while the best-performing baseline is shaded in gray (except for GATv2 using real features, which perform better than the proposed approaches for the CiteSeer dataset).

| GNN-Architecture | Trainable Node Embeddings | Deep walk Embeddings | real features |
|---|---|---|---|
| GCN | $35.4 \pm 1.3$ | $45.6 \pm 0.7$ | $66.3 \pm 1.4$ |
| GraphSage-pool | $39.7 \pm 2.9$ | $42.1 \pm 2.1$ | $64.4 \pm 3.1$ |
| GATv2 | $36.9 \pm 2.5$ | $44.4 \pm 1.9$ | $\mathbf{67.4 \pm 2.3}$ |
| GCN-*TAS* | $43.6 \pm 1.3$ | $\mathbf{47.8 \pm 1.9}$ | $66.7 \pm 0.6$ |
| GraphSage-pool-*TAS* | $46.8 \pm 2.1$ | $44.6 \pm 2.2$ | $65.3 \pm 1.5$ |
| GATv2-*TAS* | $\mathbf{48.4 \pm 3.4}$ | $46.1 \pm 0.8$ | $66.7 \pm 2.1$ |

Table 5: F1-micro Accuracies on PubMed Dataset for different initializations of features vectors for the node classification task. The best overall performance is highlighted in bold, while the best-performing baseline is shaded in gray.

| GNN-Architecture | Trainable Node Embeddings | Deep walk Embeddings | real features |
|---|---|---|---|
| GCN | $50.7 \pm 1.9$ | $70.5 \pm 1.3$ | $78.1 \pm 0.6$ |
| GraphSage-pool | $43.2 \pm 1.5$ | $70.6 \pm 2.5$ | $77.1 \pm 1.5$ |
| GATv2 | $46.7 \pm 0.8$ | $69.4 \pm 1.4$ | $78.4 \pm 1.2$ |
| GCN-*TAS* | $\mathbf{62.3 \pm 1.2}$ | $74.7 \pm 1.0$ | $\mathbf{79.6 \pm 1.3}$ |
| GraphSage-pool-*TAS* | $52.4 \pm 2.3$ | $\mathbf{76.2 \pm 1.6}$ | $79.2 \pm 0.8$ |
| GATv2-*TAS* | $54.1 \pm 3.9$ | $71.7 \pm 2.2$ | $78.2 \pm 3.1$ |

(2016a) and GraphSage Hamilton et al. (2017a) on different sampled subgraphs on the *CORA* Sen et al. (2008) and *OGBN-PRODUCTS* Hu et al. (2020)datasets, respectively, for understanding purposes. We sample 5 connected induced subgraphs for experimental purposes with 3000 vertices each. We ran the Step 2 of Algorithm 4 to find the optimal sets of parameters for each of these 5 subgraphs. We ran the GCN on Cora on a similar train-test setup as described in Section 6 and used the standard train-test splits on ogbn-products. As can be seen from Table 7, the accuracies attained for real features and trainable node embeddings are similar to the observed accuracies from Table 3. The slight improvement in GCN performance can be attributed to better generalization of *TAS- parameters* by using the subgraphs rather than the entire graph.

Table 6: F1-micro Accuracies for node classification task on Cora Dataset using GCN with *TAS sampling*. The TAS parameters for Algorithm 4, Step 2 were determined for five randomly sampled induced subgraphs of 3000 nodes (*SUBGRAPH-i*, $i \in [0, 4]$), and *TAS-Sampling* for GCN is performed on the entire graph $G$ using the optimal parameter set found on each of these sub-graphs.

| TAS Optimal set | Trainable Node Embeddings | real features |
|---|---|---|
| Subgraph-0 | $72.7 \pm 1.0$ | $77.3 \pm 1.1$ |
| Subgraph-1 | $70.6 \pm 1.4$ | $78.2 \pm 0.9$ |
| Subgraph-2 | $71.1 \pm 0.9$ | $77.7 \pm 2.8$ |
| Subgraph-3 | $71.1 \pm 2.1$ | $77.4 \pm 0.4$ |
| Subgraph-4 | $70.8 \pm 1.8$ | $77.4 \pm 2.4$ |

Table 7: F1-micro Accuracies for node classification task on ogbn-products dataset using GraphSage with *TAS sampling*. The TAS parameters for Algorithm 4, Step 2 were determined for five randomly sampled induced subgraphs of 10000 nodes ($SUBGRAPH$-$i$, $i \in [0, 4]$). *TAS-Sampling* for GraphSage (for 1000 epochs) was then performed on the full graph $G$ using the TAS optimal parameter set found on each of these sub-graphs.

| TAS OPTIMAL SET | TRAINABLE NODE EMBEDDINGS | REAL FEATURES WITH TAS SAMPLING | REAL FEATURES WITH SAGE SAMPLING |
|---|---|---|---|
| SUBGRAPH-0 | $63.1 \pm 1.7$ | $79.2 \pm 1.6$ | $77.1 \pm 2.3$ |
| SUBGRAPH-1 | $64.9 \pm 1.3$ | $78.9 \pm 1.1$ | $78.1 \pm 0.9$ |
| SUBGRAPH-2 | $62.4 \pm 1.5$ | $80.7 \pm 0.8$ | $77.9 \pm 1.3$ |
| SUBGRAPH-3 | $63.7 \pm 0.8$ | $81.4 \pm 1.7$ | $79.9 \pm 1.1$ |
| SUBGRAPH-4 | $64.5 \pm 1.0$ | $82.1 \pm 1.9$ | $80.2 \pm 0.6$ |

## 7 Conclusion

This paper introduces two novel techniques for generating node embeddings in non-attributed graphs: Maximum Adjacency Search (MAS) and Threshold-based Adjacency Search (TAS), which are inspired by the theory of $s - t$ minimum cuts. Both algorithms proved to be effective in downstream tasks. In link prediction tasks, we observed an average improvement of 6.97% over existing shallow embedding techniques. Additionally, when incorporated as neighborhood sampling mechanisms, our approaches exhibit an improvement up to 12.1% in node classification tasks compared to Graph Neural Networks equipped with artificial node features, highlighting the effectiveness of our sampler in identifying structurally influential vertices (that can be at multi-hop distance) around every node.

**Limitations** Our approaches are better suited for networks where attributes and links are predominantly based on homophily. We empirically demonstrated in Appendix, Section D, where our methods struggle to identify broker edges or bridging ties that typically form between different well-connected structures. Future research should be aimed at the node features and the connectivity structure of a graph while simultaneously adapting for both homophily and heterophily in graphs. Given that the proposed approaches require multiple runs to evaluate the optimal hyper-parameter values, future research should focus on designing models with fewer hyper-parameters.

**Impact Statement** This paper presents work whose goal is to advance the field of graph-based Machine Learning. There are many potential societal consequences of our work, none of which we feel must be specifically highlighted here.

## 8 Acknowledgments

This work was supported by the Swiss National Science Foundation (SNSF Grant number 209488).

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

# Appendix: Adjacency Search Embeddings

## A  Scalability of MAS and TAS approaches

**Why is MAS scalable to large networks:**  For each node and a subset of its neighborhood (where the subset is chosen in step 9 of Algorithm 1), MAS generates a sequence of nodes of constant length. This is done by iteratively adding a vertex that is maximally connected to the existing sequence. This process can be performed for all vertices in the graph in overall linear time, as explained in the time complexity Section 3.4 of the paper. The data fed into the common SkipGram Word2vec model has a size of $\mathcal{O}(n \times \text{walk length})$. Since the walk length is constant, the space complexity is linear in the number of nodes. The size of the generated embeddings is $\mathcal{O}(n \times d)$, where $d$ represents the dimension size.

**Why is TAS scalable to large networks:**  In TAS, each node is assigned a threshold, denoted as $\theta \in [1, 8]$. When $\theta = 1$, TAS generates sequences of constant length by conducting a BFS on the subset of nodes chosen in line 10 of Algorithm 2. When $\theta = 2$, a node outside the selected subset of nodes (line 10 of Algorithm 2) will be activated or added to the sequence of existing nodes if it has at least two edges incident to the current set of nodes. Similarly, for $\theta = i$, a node is added to the existing sequence of nodes if it is incident to at least $i$ nodes that are part of the existing sequence. This operation can be executed by slightly modifying the standard Breadth First Search, as depicted in Algorithm 3. This demonstrates that TAS is as scalable as the standard BFS, enabling it to handle extremely large-scale networks. The complexity analysis of TAS is akin to MAS, with a slightly higher constant factor, as sequences are generated for every node and a subset of its neighbors from $\theta = 1$ to 8.

## B  Statistics of Datasets Used

1. Facebook Leskovec & Mcauley (2012): This dataset captures a snapshot of users' ego networks, where nodes represent users, and edges depict their relationships.
2. Arxiv Leskovec et al. (2007): A collaboration network of researchers who submitted articles to the Astrophysics category in ArXiv. Here, nodes represent the researchers, and edges indicate collaborations on papers.
3. Email Yin et al. (2017): Constructed from email data from a European research institution, this network has an edge $(u, v)$ if person $u$ sent at least one email to person $v$. The labels represent department IDs.
4. Cora Sen et al. (2008): A citation network of publications classified into 7 classes based on their subject areas.
5. Pubmed Namata et al. (2012): A citation network of scientific publications related to diabetes, with labels denoting the type of diabetes discussed in each publication.
6. Citeseer Sen et al. (2008): A citation dataset of scientific publications where the research articles are classified into 6 subject categories.
7. Blog Zafarani & Liu (2009): A friendship network in which nodes represent bloggers and edges represent friendships between them. The labels represent blogger interests inferred through the metadata provided by the bloggers.

Assortativity, or homophily, refers to the inclination of nodes to connect with others that are similar in some aspect. Table 8 presents the degree and label assortativity, computed according to Newman (2003) on the datasets used in this paper. Email, Cora, and Pubmed exhibit negative degree assortativity, indicating that nodes in these graphs are more likely to connect to nodes with different degrees. Conversely, the graphs Facebook, Arxiv, and Citeseer demonstrate positive degree homophily, suggesting that nodes in these graphs tend to connect with others that have similar degrees. The Email dataset presents negative label assortativity, i.e., connected nodes have different labels, while others present positive homophily. Thus, the datasets chosen cover a broader spectrum of homophily properties.

Table 8: Statistics of Datasets Used

| Dataset | Nodes | Edges | Avg. Degree | # Labels | Degree Homophily | Label Homophily |
|---|---|---|---|---|---|---|
| FACEBOOK | 4,039 | 88,234 | 21.84 | NA | 0.0635 | NA |
| ARXIV | 18,772 | 198,110 | 10.55 | NA | 0.2052 | NA |
| EMAIL | 1,005 | 25,571 | 25.44 | 42 | $-0.0137$ | $-0.0029$ |
| CORA | 2,708 | 5,429 | 2.004 | 7 | $-0.0659$ | 0.7711 |
| PUBMED | 19,717 | 88,651 | 4.49 | 3 | $-0.0436$ | 0.686 |
| CITESEER | 3,312 | 4,732 | 1.42 | 6 | 0.0493 | 0.6778 |
| BLOG | 10,312 | 333,983 | 32.38 | 39 | $-0.2541$ | 0.0515 |

## C  Operators for Link Prediction Tasks

Table 9 provides the list of operators used for link prediction tasks against other shallow embedding approaches.

Table 9: Choice of operator for link prediction tasks

| Operator | Definition | symbol |
|---|---|---|
| HADAMARD | $f_i(v) * f_i(u)$ | $\odot$ |
| L2 | $\|f_i(v) - f_i(u)\|^2$ | $l2$ |
| L1 | $\|f_i(v) - f_i(u)\|$ | $l1$ |
| AVERAGE | $\|f_i(v) + f_i(u)\|$ | $\mu$ |

## D  Applicability of MAS and TAS approaches

This section delves into the operational aspects of our proposed embedding approaches. Specifically, we will showcase the network characteristics that warrant consideration before applying MAS and TAS approaches.

**Analysis on Facebook and CORA Networks:**  In the context of link prediction, we analyze the challenging nature of predicting certain network edges using MAS and TAS embeddings. We hypothesize that the misclassified edges are often connected to nodes that act as *brokers* or *bridges* between two different portions of the sub-graphs and have higher betweenness centrality Park & Neville (2023); Everett & Valente (2016); Freeman (1977); Brandes (2001). To test this hypothesis, we categorize nodes based on their betweenness centralities and assign each misclassified edge to the bucket containing the higher betweenness centrality of its two endpoints. Figures 2(a) and 2(b) depict the distribution of misclassified edges within each bucket for the Facebook and Cora networks. The 0-10 percentile group corresponds to the lowest 10 percentile of betweenness centralities, while the 90-100 percentile group represents the top 10 percentile.

Our observation reveals a skewed distribution towards the 90-100 percentile bucket for both networks, implying that misclassified edges are frequently associated with these broker nodes with higher betweenness values as one of its endpoints. These broker nodes act as bottlenecks for both MAS and TAS approaches, as the sequences can only percolate through these vertices, typically for very low values of threshold or adjacencies. Consequently, this results in a *limited* set of sequences surrounding the broker node for learning the context of the node. This observation is further illustrated for test edges with vertices of degree 1 or 2 as the sequences can only percolate with thresholds 1 or with a maximum adjacency of 1. Similar challenges arise for vertices with star-shaped topologies or for vertices connected in long chains. We observed similar trends for misclassified node labels in node classification tasks.

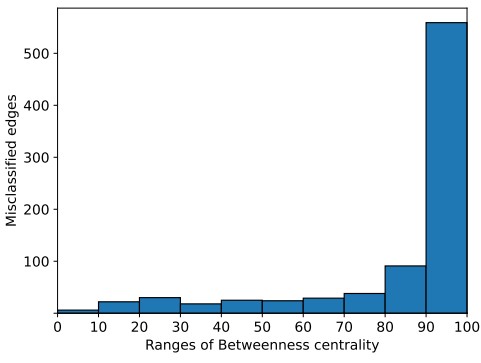 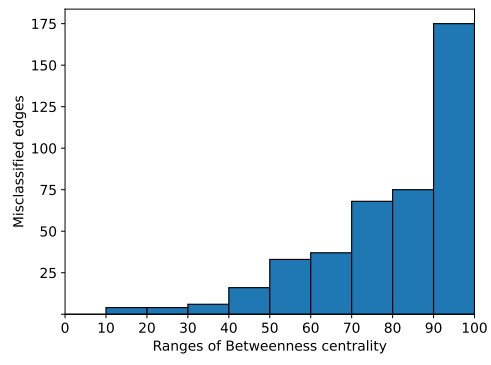

(a) Incorrectly predicted edges on Facebook.       (b) Incorrectly predicted edges on CORA.

Figure 2: Betweenness Centralities of the end points of misclassified edges with MAS

**Parameter Sensitivity:**  In assessing the impact of minor perturbations in parameters on the performance of MAS and TAS embedding approaches for prediction and classification, we first determine the optimal parameter configuration that maximizes performance. We then systematically vary each selected parameter by $\pm 1$ and observe the resulting changes in performance. For MAS, we perform experiments to assess sensitivity by perturbing the number of permutations ($p$) and the number of neighbors ($k$). For TAS, the sensitivity analysis is performed by varying the threshold.

**Sensitivity w.r.t to number of permutations**  For Maximum Adjacency Search embeddings, a slight perturbation in the number of permutations parameter $p$ yields an average variation of 1.87% and 2.03% for link prediction and node classification, respectively for Cora, Citeseer and Pubmed networks. We notice similar results when perturbing the neighbors parameter, $k$. Overall, we observe an enhancement in performance with the increase of $p \in [1, 8]$ and $k \in [1, 10]$; this is expected as the sequences are generated for diversified neighborhoods of each vertex with increase in the value of $k$. In the case of Threshold-based Adjacency Search, a minor perturbation in $\theta$ by $\pm 1$ results in an average variation of 1.5% and 3.21% for link prediction and node classification tasks. As described in Section 3, increasing the value of $\theta$ restricts the spread of Delayed-BFS (Algorithm 3) to specific higher-order structures.

**Sensitivity w.r.t walk-length**  For MAS, varying the walk-length parameter $l$ ($l \in \{5, 10, 15, 20, 25, 30, 35, 40\}$) results in an average performance variation of 2.9% for link prediction and 1.51% for node classification on the Cora, Citeseer, and Pubmed networks. Generally, performance improves as the walk length increases, which is expected since longer walks capture substructures in multi-hop neighborhoods more effectively. Similar trends in performance variation have been observed for TAS.

**Sensitivity w.r.t dimensions**  To illustrate the impact of the dimension parameter $d$ on both link prediction and node classification tasks for the MAS approach (Algorithm 1). As depicted in Figure 3(a) and Figure 3(b), the accuracy demonstrates an upward trend with an increase in dimension. Notably, in the case of Cora's link prediction, there is a significant accuracy improvement of 3.2% when varying $d$ from 128 to 512. A parallel trend is evident in the Citeseer dataset for node classification, showcasing a Micro-F1 score increase of 2.41%. The behavior of TAS-embeddings with the change in dimension is similar to that of the MAS approach.

# E  Structural Encodings

In graph neural networks (GNNs), structural encodings are often concatenated with real node or edge features to enhance the model's ability to capture both local topology and node-specific attributes. These structural encodings typically represent graph properties, such as node degrees, centrality measures, or subgraph patterns, and are used to provide additional context about a node's position and importance within the

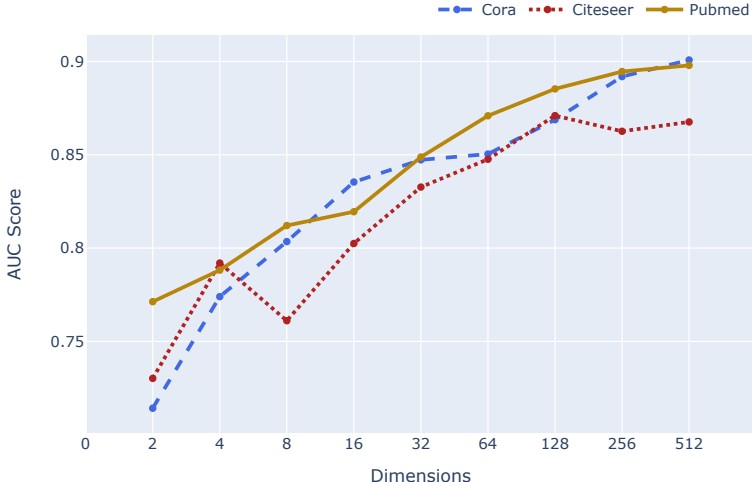

(a) Change in AUC scores for link prediction task on the Cora, Citeseer, and Pubmed datasets with the change in dimensions.

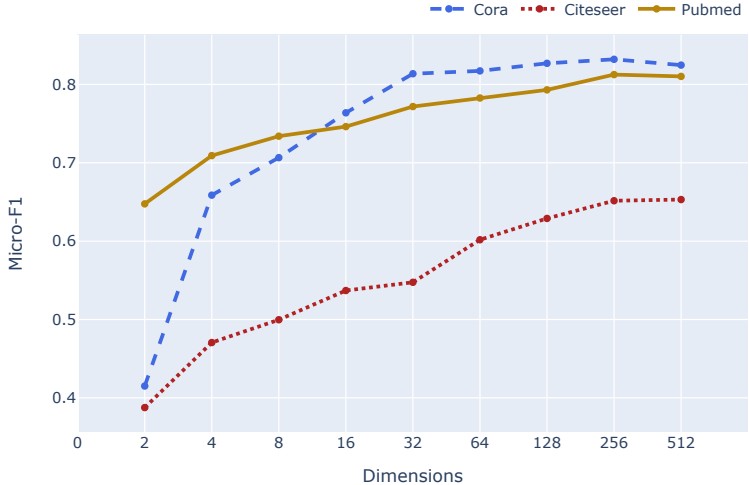

(b) Change in Micro-F1 scores for node classification tasks on the Cora, Citeseer, and Pubmed datasets with the change in dimensions.

Figure 3: Sensitivity of dimension parameter on MAS-embeddings

graph. When these encodings are concatenated with real features, the model benefits in enhanced expressivity Michel et al. (2023), capturing multi-hop information and better generalization by leveraging node's intrinsic features and their structural context.

In Table 10, we present the results of concatenating TAS embeddings, generated from Algorithm 2, with intrinsic node features for GraphSage using random sampling (similar trends were observed for GCN and GATv2). We found that TAS proves to be more efficient when used as a sampler in message passing for GNNs

(i.e., as an implicit structural encoding) compared to the explicit approach, where structural encodings are concatenated with node features. This suggests that leveraging TAS within the message-passing framework leads to better performance than directly concatenating embeddings.

Table 10: F1-micro Accuracies on Cora, Citeseer and PubMed Datasets using structural encodings for node classification task. Features vectors are concatenated with TAS embeddings and node's intrinsic features and passed it to GraphSage with random sampling at every layer.

| Graph | real features + TAS Embeddings | real features |
|---|---|---|
| Cora | **77.9 ± 0.8** | 77.7 ± 1.1 |
| Citeseer | 63.1 ± 2.7 | **64.4 ± 3.1** |
| Pubmed | **77.5 ± 0.8** | 77.1 ± 1.5 |

## F  Training Time for TAS approach

Table 11 provides the training time (in minutes($m$) and seconds($s$)) for the TAS, and other random walk based methods (Node2Vec and Deepwalk) across various parameters on node classification tasks. Training times were obtained using 16 core processor, running TAS on 12 threads, and all algorithms were implemented using gensim.

Table 11: Training times across networks

| Graph | TAS | Node2vec | DeepWalk |
|---|---|---|---|
| Email | $11m33s$ | $18m41s$ | $13m55s$ |
| Cora | $49m09s$ | $61m13s$ | $24m31s$ |
| CiteSeer | $33m47s$ | $43m09s$ | $27m16s$ |
| PubMed | $121m02s$ | $187m25s$ | $39m08s$ |

