# OpenReview forum: "Adjacency Search Embeddings"
_TMLR — Accepted by TMLR_

### Review · Reviewer_z7HS · 2024-08-03

**Summary Of Contributions:**

This paper aims to propose novel and better node embedding for graph learning. Specifically, two methods are proposed, including MAS and TAS. MAS is inspired by the Maximum/Cardinality search technique, which is designed to identify the s-t minimum cut on graphs. TAS is a generalised variant of MAS, in which a threshold is introduced.

The proposed method is tested on Cora and Citeseer datasets, and is compared to several baselines.

**Audience:**

Yes

**Broader Impact Concerns:**

No ethical concern

**Claims And Evidence:**

Yes

**Requested Changes:**

Please refer to the weakness part

**Strengths And Weaknesses:**

Strengths:

This paper studies the important problem of graph embedding and proposes two novel embedding methods.

Weaknesses:

The works reviewed in intro are not up to date (most are earlier than 2020) and lack the works proposed in recent years

What is the motivation of the proposed two methods. Why is the proposed methods more advantageous than existing works? It is mentioned that the proposed algorithm can better capture some types of graph motifs. However, it seems that these properties have also been studied in previous works and are not unique to the proposed method in this work.

Please explain more on the permutation. what is the meaning and how does it influence the empirical performance.

The adopted baselines are old, it would be better if the proposed method can be compared against recent state-of-the-art methods.

Please explain what does it mean by ‘maximally adjacent’. For example, in the algorithm of MAS, it is used to decide a node that is maximally adjacent to a set of nodes.

Please explain more on the s-t minimum cut. This part is also relevant to the motivation of the proposed method but is not clearly explained. It is unclear why the techniques designed for s-t minimum cut is advantageous to be adopted for node embeddings.

It would be better if larger datasets with more diverse properties can be adopted to tested the proposed method. Currently, the adopted Cora and Citeseer are small toy datasets and are all citation networks. Therefore it is unclear whether the proposed method is generally applicable.

---

> ### Author Response · Authors · 2024-10-13
> **Rebuttal**
>
> We thank the reviewer for taking the time to provide valuable feedback.
>
> Below, we address each of the concerns raised by the reviewer:
>
> **Literature review of works proposed in recent years**: We will incorporate additional path-based and higher-order information-based measures, as recommended by Reviewer YuEp. Furthermore, we will explain how our methods differ from these existing measures.
>
> **Motivation of the proposed approaches*: Our approaches are motivated by Maximum Adjacency Search (MAS), which is a well-known $s$-$t$ minimum cut approach. They are built on the assumption that the probability of future connections between two different nodes is higher in well-connected structures.
>
> Thus, they are suited for networks where attributes and links are shared based on homophily. This has been empirically demonstrated in Appendix, section D, where our approaches fail to identify the broker edges or bridging ties that usually arise between different well-connected structures.
>
> *Advantages compared to existing methods*: Unlike random walks, which sample sequences from every node, our approaches generate specific sequences that capture well-connected local neighborhoods surrounding every node. This reduces training time as the number of sequences generated is lower than the random walk counterparts. Unlike existing higher-order approaches, our methods identify these well-connected  regions without explicitly looking for combinatorial structures such as cliques, etc.
>
> **How does permutation influence the empirical performance.**:
> Permutation parameter helps us to capture well-connected nodes connected to different subsets of neighborhoods for each node. Empirically, we observed that as the number of permutations increases, cohesive structures (sequences) across different diversified neighborhoods are captured, resulting in increased performance.
>
> **Baselines Used**: Our objective was to evaluate the performance of our technique in comparison to leading shallow embedding methods while also extending these approaches to GNNs. To achieve this, we conducted experiments on several datasets, including Cora, Pubmed, Citeseer, Facebook, arXiv, Email, and OGBN-PRODUCTS. The size of the datasets ranged from 1,005 to 2,449,029 nodes. In Appendix Section D, we discussed the applicability of our methods. They can be used as a graph sparsification mechanism for graphs where attributes and links are shared based on homophily. We will make this explicit in our write-up.
>
> Additionally, for the rebuttal, we incorporated our method into existing graph transformer architecture as context nodes for positional encodings and compared it with NAGPHORMER (ICLR 2023). NAGPHORMER constructs a sequence for each node based on the tokens of different hops of neighbors. In comparison, we use the TAS approach to create sequences for each node based on the tokens of influential/structural-aware neighborhoods. We observed a performance improvement of up to 2.3% on the CORA network. This experiment is only to demonstrate the efficacy and applicability of our approaches in other GNN architectures. An in-depth comparison across all transformer architectures is beyond the scope of this article and will be pursued in our further study.
>
> **Maximal Adjacency in MAS**: In MAS, maximal adjacency refers to the vertex or node with the highest (weighted) number of edges connected to the nodes already included in the generated sequence.
>
> **$s$-$t$ minimum cut approach**: MAS is a recursive algorithm where, at each iteration, the objective is to identify a tightly connected vertex to the set of well-connected vertices and perform a merge operation to determine the s-t minimum cut. *Incorporating the vertex with the highest degree of connectivity into the existing set of vertices reduces the likelihood of encountering an edge belonging to the minimum cut*. Thus, this method is highly likely to capture the well-connected regions surrounding every node.
>
> We are happy to add all the above clarifications in our final version.
>
> We hope to have answered all your questions. Please let us know if you have any further questions.

---

> > ### Comment · Reviewer_z7HS · 2024-10-25
> >
> > Thanks for the detailed response from the authors. The authors have also uploaded a revised version. The concerns are resolved now.

---

### Review · Reviewer_YuEp · 2024-09-03

**Summary Of Contributions:**

This paper proposes two new embedding methods of the adjacency matrix: Maximum Adjacency Search (MAS) and Threshold-based Adjacency Search (TAS). The idea of the methods is to generate higher-order representations of nodes from the graph structure.

The proposed methods are shown to outperform existing shallow embedding methods like Node2Vec and DeepWalk in tasks such as link prediction and node classification.

The paper also discusses the use of MAS and TAS as preprocessing techniques to improve the performance of Graph Neural Networks (GNNs) such as GCN, GraphSage, and GATv2.

**Audience:**

Yes

**Broader Impact Concerns:**

I have no concerns.

**Claims And Evidence:**

Yes

**Requested Changes:**

Please refer to the weaknesses section in my review. In particular, I think that it is important that you address points 1,2, and 4. Points 3 and 5 are very important and interesting in my opinion, but 1,2, and 4 are more core components I would expect to see in the final paper.

**Strengths And Weaknesses:**

**Strengths:**

1. To the best of my knowledge, the proposed MAS and TAS are new, and are different than existing random walk based methods.

2.  The experimental results demonstrate improvements in link prediction and node classification tasks compared to other shallow embedding techniques, and good performance as input features for GNNs.

4. The paper evaluates the applicability MAS and TAS on both attributed and non-attributed graphs, showing their relevance to a different applications.

5.  The authors show how MAS and TAS can be combined into GNNs, thereby going beyond simple shallow embedding techniques.

**Weaknesses:**
1. Missing discussion of some related works, in particular:

1A. The relation to methods that pre-compute higher order information on the graph, such as [1] and [2].\
1B. The relation to other 'path' and sampled based methods in GNNs like [3,4,5].

2. The performance of MAS and TAS is  likely sensitive to its hyperparameters. For example, the number of permutations and the walk length are crucial factors of such methods.  I would highly recommend that the authors add an experiment that shows the sensitivity of the method to its hyperparameters.

3. Results on synthetic graph substructure benchmarks (such as shown in [2], and [6] that is cited in the paper) could help to shed light on the expressive power that can potentially be added by the proposed method.

4. Results on datasets **with** node features would make the experiments more convincing.For example, the authors show results on Cora, Citeseer, and Pubmed, but to my understanding they do not consider the case where both the embeddings computed by MAS and TAS are combined with input node features. I think that it would be beneficial to show if there is merit in combining the two, as these essentially can be regarded as sort of Structural Encoding, and these are known to often times help performance wise.

5. Because some elements of the method are stochastic, to the best of my understanding, it would be interesting to know how stable it is. i.e., how much do the results fluctuate between different realisations of the embeddings through different sampling over the graph? does the method converge to something if we take infinitely many samples?

[1] Improving Graph Neural Network Expressivity via Subgraph Isomorphism Counting

[2] An Efficient Subgraph GNN with Provable Substructure Counting Power

[3] pathGCN: Learning General Graph Spatial Operators from Paths

[4] Shortest Path Networks for Graph Property Prediction

[5] Path neural networks: expressive and accurate graph neural networks

[6] Random Features Strengthen Graph Neural Networks


**Minors:**
-  A typo in the introduction,  "These shallow embedding techniques, though computationally efficient, suffers from the drawback of lacking parameter sharing during training." – "suffers" should be "suffer."
-  Grammatical error, "This approach draws inspiration from the Maximum Adjacency/Cardinality search technique, which is employed for identifying s-t minimum cut in a graph." – Should be "an s-t minimum cut."

---

> ### Author Response · Authors · 2024-10-13
> **Rebuttal**
>
> We thank the reviewer for taking the time to provide valuable feedback.
>
> Below, we address each of the concerns raised by the reviewer:
>
> **Missing discussion of some related works**: We will make sure to add discussion in related work sections and also add a paragraph comparing how our methods differ in the final version of our paper.
>
> **Sensitivity to its hyperparameters such as number of permutations and walklength**:  MAS and TAS are indeed sensitive to the parameter "number of permutations." This parameter allows us to identify well-connected nodes linked to different subsets of neighborhoods for each node. Empirically, we observed that increasing the number of permutations captures more cohesive structures (sequences) across diverse neighborhoods, leading to improved performance. We have experimental results that explore the impact of varying both the walk length and the number of permutations, and we will include these findings in the sensitivity analysis section related to hyperparameters in the final version of our paper.
>
> **Structural Encodings**:  We did experiments by concatenating the embeddings produced by MAS/ TAS approaches with the real features, and interestingly, we did not see an increase in accuracies by incorporating the structural encoding when compared to the real features. For Cora, the real features with TAS sampling perform better by 1.1% compared to the concatenated embeddings (real + TAS embeddings). Similar results are observed for Pubmed and Citeseer networks. We will incorporate our findings in the final version of our paper.
>
> We will also aim to incorporate points (3) by additional substructure comparisons and (5) by an empirical assessment of how much the results fluctuate between different realizations of the embeddings through different sampling over the graph in our final version.
>
> We thank the reviewer for pointing out the typos. We will address them in the final version.
>
> We hope to have answered all your questions. Please let us know if you have any further questions.

---

> > ### Comment · Reviewer_YuEp · 2024-10-14
> > **Discussion**
> >
> > Thank you authors for the rebuttal.
> >
> > Just to clarify one thing which I am not fully sure about - in the experiments you show in the current version. Did you use node features on datasets like Cora or not ? From the text it seemed like you do not, but now your explanation sounds like you did. Please clarify this both here and in the paper. If I understand your comment now, you say that to produce MAS/TAS you **use the original features**, but then it is not necessary to somehow include them again in addition to the MAS/TAS because they are already a function of the input features? If so, that is fine, but I think that it should be made clear in the paper and also showing the results you described in your rebuttal are important.
> >
> > To conclude, the promised changed sound adequate to me - but I would like to see the revised version.
> >
> >
> > Thank you.

---

> ### Author Response · Authors · 2024-10-14
>
> Dear Reviewer,
>
> Thank you for your question.
>
> In the fourth column of the Tables 3,4,5 of the paper, we experimented with the real node features for Cora, Citeseer, and Pubmed datasets.  So, we did experiments with non-attributes (random initialization) and real- attributes in the under-review (submitted) version of the paper.
>
> *Clarification on Structural Encodings*: MAS and TAS approaches described in the shallow embedding section (Tables 1 and 2) do not use real-node features (they only use the structure/topology of the network). In the experiments conducted in the rebuttal period, we concatenated the embeddings generated by MAS/TAS approaches (which can be considered as structural embeddings) with the real node features and passed them to GNNs (with TAS sampling). There, we did not notice any performance improvement (as they are already a function of aggregation of input node features (due to TAS sampling) in GNNs).
>
> We will make this clear by adding these results in our final version of the paper.
>
> We hope to have answered your questions. Please let us know if you have any further questions.

---

> > ### Comment · Reviewer_YuEp · 2024-10-14
> > **Thank you**
> >
> > Thank you for the clarifications. I look forward to read the revised version.

---

> > > ### Author Response · Authors · 2024-10-14
> > > **Revised Version**
> > >
> > > We will upload the revised version with all the changes by the end of this week. We thank you again for your valuable time.
> > >
> > > Thank you,
> > > Authors

---

> ### Author Response · Authors · 2024-10-20
> **revision with changes**
>
> Dear Reviewer,
>
> We have uploaded the revised version of our manuscript.
>
> We addressed your points (1), (2), and (4) raised during the rebuttal.
>
> (1) Added related works and their comparisons in section 5.1  (2nd and 3rd paragraphs)
>
> (2) Sensitivity to hyperparameters: number of permutations and walk length are included
>
> (4)  Results for structural encodings (concatenated TAS embeddings and Real features) with random sampling are provided in Appendix E.
>
> Please let us know if you want us to include any further information. We thank you again for your valuable time.

---

> > ### Comment · Reviewer_YuEp · 2024-10-25
> > **Thank you**
> >
> > I thank the authors for the detailed rebuttal and manuscript revision. I have no further concerns.

---

### Review · Reviewer_scgH · 2024-10-01

**Summary Of Contributions:**

This paper introduces two novel methods for generating node embeddings in graph structures, namely Maximum Adjacency Search (MAS) and Threshold-based Adjacency Search (TAS). These methods are inspired by the theory of s-t minimum cuts and aim to capture well-connected nodes within a neighborhood, which are essential for generating higher-order representations. This node embedding approach integrates the Skip-Gram model for both link prediction and node classification tasks and also serves as a pre-processing technique for Graph Neural Networks (GNNs).

**Audience:**

Yes

**Broader Impact Concerns:**

The broader impact section is not included in this article, and based on my assessment, there are no concerns regarding the ethical implications of this work that need to be addressed.

**Claims And Evidence:**

No

**Requested Changes:**

Please refer to the above-mentioned weaknesses to make revisions to this article.

1.The revised version should directly address the various limitations identified in the proposed approaches.
2.It should clearly articulate the motivation for and advantages of the proposed approaches.
3.Please conduct a thorough review of the literature closely related to the proposed methods.
4.Extensive experiments should be performed to convincingly demonstrate the effectiveness of the proposed approaches.
5.The organization of the paper should be rearranged to enhance readability.
6.Please clarify the descriptions of the methodology to improve understanding.
7.Ensure that all raised questions are answered thoroughly.

**Strengths And Weaknesses:**

Strengths:
1.New Node Embedding Techniques. MAS dynamically selects maximally connected nodes, while TAS uses a threshold-based approach to find cohesive substructures. These methods are used to pre-process graph data and can be integrated with models like the Skip-Gram to enhance tasks such as link prediction and node classification.
2.Performance Improvement. MAS and TAS embeddings outperform traditional shallow embeddings like Node2Vec and DeepWalk in link prediction and node classification tasks.
3.Applications in GNNs. When used as a pre-processing technique for GNNs, MAS and TAS improve the performance by better identifying influential local neighborhoods for message passing.

Weaknesses:
1.Several Limitations. One major concern arises from several limitations in the proposed approaches. (1) The primary strategy employed, Maximum Adjacency Search (MAS), tends to prioritize tightly connected neighbors, which can lead to the exclusion of sparsely connected nodes that might carry important information. This approach assumes that the most informative nodes are the ones densely connected to the current sequence. However, in certain data structures, especially in scenarios with heterogeneous or long-range connections, less connected nodes could be pivotal for capturing the broader graph structure or important latent relationships. A possible limitation of the proposed node embedding approach is its bias toward locally dense subgraphs, which can result in an incomplete or biased representation of the global graph structure. This limitation could affect tasks that require capturing weak but semantically meaningful connections, such as in social networks or knowledge graphs, where weaker links might represent important relationships. (2) The large number of parameters involved in both MAS and TAS increases the risk of overfitting, which can degrade generalization performance and limit their effectiveness in real-world applications. This over-parameterization can also lead to increased computational costs and make model tuning more challenging. (3) The computational complexity presented in this paper is inaccurate, as it assumes the graph is sparse, representing a best-case scenario. This assumption is unrealistic, as real-world graphs are more likely to be dense. When analyzing computational complexity, it is essential to consider the worst-case scenario, where the graph is dense. Moreover, the computational complexity of the proposed approaches depends on both the number of nodes and edges, which could significantly limit their scalability and applicability in real-world scenarios involving large and dense graphs.

2.Unclear Motivation and Methodological Advantages. Another major concern is the unclear motivation and advantages behind the proposed approaches. The specific research problems this paper aims to address, as well as the advantages of the proposed methods over existing ones, are not clearly defined. Consequently, it is difficult to determine whether the observed performance improvements in the experiments are due to the proposed design or simply the result of careful parameter tuning. To strengthen the paper, a clearer articulation of the research problems compared to existing methods is needed, along with concrete examples demonstrating how the proposed approaches outperform existing techniques, including node embedding methods, neighborhood sampling strategies in GNNs, and methods for extracting higher-order relationships.

3.Incomplete Literature Review. While the section on Related Work categorizes the literature into three main areas—matrix factorization, random walks, and GNNs—it remains incomplete and lacks specificity. Given that the proposed approaches focus on mining high-order relationships and performing neighborhood sampling for node embedding learning, the paper should provide a more comprehensive review of relevant studies. This should particularly include an in-depth examination of node embedding methods, neighborhood sampling techniques in GNNs, and the exploration of higher-order relationships, such as those found in hypergraphs and subgraph mining.

4.Unconvincing Experimental Evaluation. Although various experiments were conducted across different tasks, such as link prediction and node classification, the experimental evaluation falls short in convincingly demonstrating the effectiveness of the proposed approaches. First, the comparison methods are outdated and not representative of the current state-of-the-art. Second, the datasets used for different tasks lack consistency. Third, each task is evaluated using only a single metric, limiting the comprehensiveness of the assessment. Fourth, as previously mentioned, the numerous parameters involved in the proposed approaches raise questions about whether the performance improvements stem from the proposed design or merely from careful parameter tuning. Finally, the running time for all methods should be reported to ensure that the computational complexity of the proposed approaches remains within reasonable bounds.

5.Chaotic Organization. The paper suffers from several organizational issues, with explanations of parameters and algorithms scattered throughout. To improve readability, these sections should be rearranged. For instance, the parameter settings on page 4 should be moved to the experimental section. On page 6, Section 3.3, titled “Shallow Embedding Learning Framework” should be positioned closer to Section 3.1, as Algorithm 1 in Section 3.1 outlines the steps of embedding learning related to the Skip-gram model discussed in Section 3.3. Additionally, the discussion on neighborhood sampling found on page 9 should be incorporated into Section 2, titled “Related Work.” Moreover, an independent section should be added to define the notations used in this paper in detail. Importantly, including a separate “Preliminaries” section to introduce the techniques employed—such as Maximum Adjacency Search (MAS) and s-t minimum cut—would provide valuable context for readers.

6.Unclear Descriptions. The presentation of algorithms in this paper is difficult to understand and lacks sufficient explanation regarding the steps involved in the proposed algorithms. For example, in Algorithm 1, it is unclear what “neighbors ← permute(v.neighbors())” signifies and which technique is employed to permute the neighbors of the nodes. Additionally, the expression “S.insert(neighbors[l · k...l · (k + 1)])” is ambiguous and needs clarification. Furthermore, the line “f = train(w, d, sequences)” appears to indicate the embedding learning step, but it is not adequately explained within the context of this algorithm. Moreover, in the experimental section, it would be helpful to highlight that the second optimal performance serves as a benchmark for comparison. Finally, the terms “SUBGRPH-0,” “SUBGRPH-1,” and so on, in Tables 6 and 7 should be clearly defined to enhance understanding.

7.Questions. The first question is: how is the "high-order topology" referenced in Section 1 reflected in the proposed approaches? The second question is: what evidence supports the claim made in Section 1 that "our path-based approaches differ from random walk methods and can be considered specific instances of k-th order random walks for any value of k"?

---

> ### Author Response · Authors · 2024-10-14
> **Rebuttal**
>
> We thank the reviewer for taking the time to provide valuable feedback.
>
> Below, we address each of the concerns raised by the reviewer:
>
> **Limitations**: Our approaches are better suited for networks where attributes and links are predominantly based on homophily. We empirically demonstrated in Appendix, Section D, where our methods struggle to identify broker edges or bridging ties that typically form between different *well-connected* structures. We will emphasize this limitation and address it by adding a dedicated limitations section to the final version of our paper.
>
> *Hyper-parameters*: We do not think that the number of hyperparameters employed (Threshold $\theta$, Walk Length $l$, Window size $w$, Dimension $d$, #Neighbors $k$, #Permutations $p$) by our approach are higher (we are enumerating these parameters to choose the best ones - and further all these parameters are small constants, thus asymptotically they don't increase the time complexity). However, as suggested by Reviewer YuEp, we will provide sensitivity analysis concerning hyperparameters such as Permutations, Walklength, etc.
>
> *Sparse vs. Dense graphs*: We disagree with the reviewer's comment that most of the real-world large graphs are dense. We are happy to change our analysis for computational complexity so that it's suited for dense networks and not just sparse networks. For dense graphs, the time complexity becomes linear in the size of the graph for identifying the influential neighborhoods, and we will modify the later analysis accordingly in our final version.
>
> **Motivation**:  Our approaches are built on the assumption that the probability of future connections between two different nodes is higher in \emph{well-connected} structures.
>
> Our approaches are motivated by Maximum Adjacency Search (MAS), which is a well-known $s$-$t$ minimum cut approach.
>
> *Advantages compared to existing methods*: Unlike random walks, which sample sequences from every node, our approaches generate specific sequences that capture \emph{well-connected} local neighborhoods surrounding every node. This leads to the reduction of training time as the number of sequences generated is lower than the random walk counterparts. Unlike existing higher-order approaches, our methods identify these \emph{well-connected} regions **without** explicitly looking for these combinatorial structures such as cliques, etc.
>
> We will make this description explicit and add the motivation in the final version of our paper.
>
> **Literature Review**: Since the focus of the paper is not on hypergraphs, we did not mention them. However, if the reviewer finds them relevant, we are happy to. We include five additional papers, as suggested by Reviewer YuEp, pertinent to subgraph mining in our final version.
>
> **Experimental Evaluation**: To the best of our knowledge, we followed the standard datasets used in practice for these experiments. We used the standard evaluation metrics that are predominantly used in the current literature. We are happy to include any further metrics that the reviewer might suggest in the final version of our paper.
>
> *Comparision methods*: Additionally, for the rebuttal, we incorporated our method into existing graph transformer architecture as context nodes for positional encodings and compared it with NAGPHORMER (ICLR 2023). NAGPHORMER  constructs a sequence for each node based on the tokens of different hops of neighbors. In comparison, we use the TAS approach to create sequences for each node based on the tokens of influential/structural-aware neighborhoods. We observed a performance improvement of up to 2.3% on the CORA network. This experiment is only to demonstrate the efficacy and applicability of our approaches in other GNN architectures. An in-depth comparison across all transformer architectures is beyond the scope of this article and will be pursued in our further study.
>
> *Datasets for different tasks*: For the shallow embedding approaches, the datasets used for node classification are a subset of those used for link prediction, as not all networks include node labels. For GNNs, we used the same datasets commonly referenced in the literature, along with the OBGN datasets. We are happy to include experimental results with the Email dataset (1005 nodes) for GNNs, in the final version of our paper.
>
> **Organization**:  We will re-organize the paper to further enhance readability. We believe that the organization of "Shallow Embedding Learning Framework" is properly placed as MAS and TAS are also employed in GNNs (as shown later)
>
> **Descriptions**: We followed the standard notation used in the existing literature. We are happy to add sufficient explanations to the descriptions to enhance the readability.

---

> > ### Author Response · Authors · 2024-10-14
> > **Rebuttal**
> >
> > **Questions on Higher-order topology**: The nodes that are added to the sequences by MAS and TAS are part of structures such as cliques, triads, quads, etc (thus, the sequences capture polyadic relations between vertices for each node).
> >
> > A higher-order random walk typically retains some memory of previous steps, as opposed to a first-order Markov process. While MAS is not directly a memory-based walk in the strictest sense, its choice of nodes (based on maximum adjacency) implicitly reflects past choices (since it aims to explore unvisited vertices of the graph). This makes it behave similarly to a higher-order random walk, where past decisions influence the current selection (similar case with TAS). We are happy to provide further clarifications regarding this in the paper.
> >
> > We hope to have clarified all the questions. Please let us know if you have any further questions.

---

> > > ### Comment · Reviewer_scgH · 2024-11-01
> > > **Thank you for the response**
> > >
> > > Thank you for addressing most of my concerns. I would like to seek clarification on a few remaining points:
> > >
> > > 1. Limitations: (1) Hyper-parameters: The computational complexity associated with multiple hyper-parameters is a potential limitation. Since your approaches require multiple runs to evaluate all possible parameter values and select the optimal ones, numerous hyper-parameters can indeed lead to high computational overhead. I recommend mentioning this as a limitation in the "Limitations" section and considering the design of more concise models in future research. (2) Sparse vs. Dense Graphs: While real-world large graphs are often sparse, it would be valuable to consider worst-case scenarios, specifically for dense graphs, in your theoretical analysis of computational complexity. Including both best-case (sparse) and worst-case (dense) analyses is highly encouraged.
> > >
> > > 2. Experimental Evaluation: I appreciate the additional experiments. However, I believe that using only F1-micro accuracy as an evaluation metric is insufficient. Including running time in the experiments is also essential, as it would empirically validate the theoretical analysis of computational complexity.
> > >
> > > 3. Descriptions: In the experimental section, it would be helpful to explicitly highlight that the second-best performance is used as a benchmark for comparison.

---

> ### Author Response · Authors · 2024-11-04
> **Changes to the manuscript**
>
> Dear Reviewer,
>
> Thank you for taking the time to go through our rebuttal.
>
> We made the following changes to our manuscript:
>
> 1) Added limitations pertinent to hyper-parameters in the limitations section.
>
> 2) Our running time is expressed in terms of n AND m so that it covers all cases: sparse, m in Theta(n), and dense, m in Theta(n^2).
>
> 3) The training time for TAS over the networks employed is provided in Appendix Section F.
>
> 4) We highlighted the best-performing baseline in the tables, and we also explicitly mentioned the same in our captions and text.
>
> We hope to have clarified all the questions. Please let us know if you have any further questions.

---

### Decision · Action_Editor_PNZP · 2024-12-02

**Recommendation:** Accept as is

**Comment:**

The authors have proposed two new methods for node embedding, based on s-t minimum cuts. These methods use search-based algorithms rather than random walks to capture well-connected nodes within a neighborhood. These methods can be combined with methods such as skip-gram to obtain higher-order representations. The authors show improvement over shallow representations, and show that the the neighborhoods can be used to improve GNNs.

The reviewers and I agree that this work is valuable to the community -- the approaches are straightforward but show clear benefits (that are well-motivated by the authors). The reviewers had a number of concerns about the initial draft, including lack of motivation and discussion of applicability, and insufficient literature review, but these issues were resolved following discussion between authors and reviewers and updating of the manuscript. All reviewers are happy with the revised manuscript, and I recommend accepting as-is.

**Audience:**

This would be of interest to many who work with node embeddings

**Claims And Evidence:**

The claims are well-supported by the experimental evidence